# *Lobelia chinensis* Extract and Its Active Compound, Diosmetin, Improve Atopic Dermatitis by Reinforcing Skin Barrier Function through SPINK5/LEKTI Regulation

**DOI:** 10.3390/ijms23158687

**Published:** 2022-08-04

**Authors:** No-June Park, Beom-Geun Jo, Sim-Kyu Bong, Sang-a Park, Sullim Lee, Yong Kee Kim, Min Hye Yang, Su-Nam Kim

**Affiliations:** 1Natural Products Research Institute, Korea Institute of Science and Technology (KIST), Gangneung 25451, Korea; 2Division of Bio-Medical Science and Technology, KIST School, University of Science and Technology, Seoul 02792, Korea; 3College of Pharmacy, Pusan National University, Busan 46241, Korea; 4Department of Life Science, College of Bio-Nano Technology, Gachon University, Seongnam 13120, Korea; 5College of Pharmacy, Sookmyung Women’s University, Seoul 04310, Korea

**Keywords:** SPINK5/LEKTI, skin barrier, atopic dermatitis (AD), *Lobelia chinensis*, diosmetin

## Abstract

The skin acts as a mechanical barrier that protects the body from the exterior environment, and skin barrier function is attributed to the stratum corneum (SC), which is composed of keratinocytes and skin lipids. Skin barrier homeostasis is maintained by a delicate balance between the differentiation and exfoliation of keratinocytes, and keratinocyte desquamation is regulated by members of the serine protease kalikrein (KLK) family and their endogenous inhibitor SPINK5/LEKTI (serine protease inhibitor Kazal type 5/lympho-epithelial Kazal-type-related inhibitor). Furthermore, SPINK5/LEKTI deficiency is involved in impaired skin barrier function caused by KLK over-activation. We sought to determine whether increased SPINK5/LEKTI expression ameliorates atopic dermatitis (AD) by strengthening skin barrier function using the ethanol extract of *Lobelia chinensis* (LCE) and its active compound, diosmetin, by treating human keratinocytes with UVB and using a DNCB-induced murine model of atopic dermatitis. LCE or diosmetin dose-dependently increased the transcriptional activation of SPINK5 promoter and prevented DNCB-induced skin barrier damage by modulating events downstream of SPINK5, that is, KLK, PAR2 (protease activated receptor 2), and TSLP (thymic stromal lymphopoietin). LCE or diosmetin normalized immune response in DNCB treated SKH-1 hairless mice as determined by reductions in serum immunoglobulin E and interleukin-4 levels and numbers of lesion-infiltrating mast cells. Our results suggest that LCE and diosmetin are good candidates for the treatment of skin barrier-disrupting diseases such as Netherton syndrome or AD, and that they do so by regulating SPINK5/LEKTI.

## 1. Introduction

Skin mechanically protects the body from foreign substances and antigens and pre-vents loss of moisture, protein, and electrolytes [1]. The barrier function of skin is provided by the stratum corneum (SC), which is composed of densely packed keratinocytes surrounded by skin lipids (cholesterols, ceramides, and free fatty acids) [2]. Maintaining the SC layer is critical for sustaining skin barrier function [3], and skin barrier homeostasis is regulated by a fine balance between keratinocyte differentiation and exfoliation [4]. Skin exfoliation is caused by the decomposition of corneodesmosomes that fix keratinocytes in the epidermis via an intermediate filament network formed by serine proteolytic enzymes called kallikreins (KLKs) [3]. Desmosomes are specialized adhesive protein complexes located in intercellular junctions that connect cells [5]. Furthermore, fragments of desmocollin-1 (DSC1; a type of desmosomal cadherin) accumulate in the SC layer of normal epidermis, and the proteolysis of DSC1 is diminished in the SC of dry skin [6,7]. Filaggrin (filament-aggregating protein) is produced by hydrolysis from a large profilaggrin precursor protein during the terminal differentiation of epidermal keratinocytes and undergoes further proteolysis in the upper SC layer to release free amino acids, which retain moisture and support skin barrier functions [8]. Surprisingly, despite the importance of skin barrier functions, the mechanisms responsible for these functions are poorly understood.

SPINK5 (serine protease inhibitor Kazal type 5), also known as LEKTI (lympho-epithelial Kazal-type-related inhibitor), plays an essential role in skin barrier function. Defects in the SPINK5 gene lead to Netherton Syndrome, an uncommon autosomal recessive disease characterized by severe ichthyosis, atopic dermatitis (AD), congenital thyroid dysfunction, and bamboo-like hair symptoms. AD has been associated with polymorphism of the SPINK5 gene [9]. People affected by Netherton syndrome exhibit a wide range of chronic and severe skin allergic inflammations, such as AD with elevated serum immunoglobulin E (IgE) levels [10,11], and causes dehydration, mucosal epithelial infections, hyperlipidemia, hypothermia, and dramatic weight loss, which often lead to postnatal death due to severe changes in skin barrier function [12]. In normal epidermis, barrier function is maintained by tight cell-to-cell binding in the SC. Exfoliation involves loss of superficial keratinocytes due to epidermal protease-induced cornoedesmosome decomposition [13]. Reduced expression or activity of SPINK5 in Netherton syndrome increases the activities of serine proteases that impair cohesion in the SC, which leads to SC thinning, delayed skin repair, and dehydration and has fatal consequences for fetuses [14,15].

SPINK5 deficiency in skin accelerates degradations of desmoglein 1 (DSG1) and DSC1, which connect keratinocytes, and is caused by over-activation of KLK7 (an SCCE (stratum corneum chymotryptic enzyme)) and KLK5 (an SCTE (stratum corneum tryptic enzyme)). As a result, skin peeling occurs due to weakened cell-to-cell adhesion in the SC layer, and barrier function is lost [16]. Under these conditions, SPINK5 selectively inhibits KLK5, KLK7, and KLK14 in the epidermis to control exfoliation [17]. In addition, KLK5 deficiency induced by SPINK5 induces inflammatory and allergic reactions despite the absence of environmental factors. SPINK5 deficiency in keratinocytes directly causes the over-activation of KLK5 and the successive activations of PAR2 (protease-activated receptor 2), TSLP (thymic stromal lymphopoietin), and tumor necrosis factor α (TNF-α). This dysregulation of KLK5 breaks down the skin barrier by destroying glia between the stratum granulosum (SG) and the SC, and results in the secretion of chemical messengers, such as TARC (thymus and activation-regulated chemokine) and MDC (macrophage-derived chemokine), which produce a Th2 environment and activate eosinophils and mast cells [18,19]. Collectively, SPINK5 deficiency induces a pro-Th2 microenvironment in which inflammatory response-activated keratinocytes, eosinophils, and mast cells promote a phenotype resembling that of AD [20]

The major components of *Lobelia chinensis* extracts are apigenin 7-O-rutinoside, dios-metin, diosmin, linarin, lobeline, lobelanidine, lobelanine, lobetyolin, luteolin, nor-lobelanine, radicamine A, radicamine B, lobechidine, and propyllobelionon [21,22,23,24]. Stud-ies conducted over past decades have demonstrated that *L. chinensis* extracts have several physiological effects, which include anti-inflammatory effects on components of the NF-κB pathway and anti-proliferative effects on arterial smooth muscle and cancer cells, in addition to anti-bacterial, anti-venom, anti-hyperlipidemia, diuretic, and vascular en-dothelial protective effects [25,26,27,28].

Recent research on SPINK5 has focused on polymorphisms of the SPINK5 gene and its effects on the clinical symptoms of Netherton syndrome, and relatively few studies have been conducted on natural products or single compounds that modulate or increase SPINK5 expression. In a previous study, we found that increasing the expression of SPINK 5 using compound K in a UVB-induced photo-aging model or a DNCB-induced AD model improved skin barrier function and ameliorated atopic symptoms [29]. Therefore, in this study, we investigated whether the ethanol extract of *L. chinensis* (LCE) or diosmetin (an active compound isolated from LCE) increases the expression of SPINK5 and ameliorates the symptoms of AD by strengthening skin barrier function.

## 2. Results

### 2.1. Effects of LCE on Impaired Skin Barrier Function and Serum Factors in the DNCB-Induced AD Model

To investigate the effects of SPINK5 on skin, we first confirmed that LCE dose-dependently enhances the transcriptional activity of *SPINK5* promoter (Appendix A). Phenotypic changes were evaluated by topically treating mice with DNCB or DNCB plus LCE. Erythema, dryness, and hyperkeratosis were observed in DNCB-induced AD mice, but these features were ameliorated by LCE treatment (Figure 1A). Skin barrier function is maintained by a balance between water loss and retention and skin surface pH. Transepidermal water loss (TEWL) provides a measure of skin barrier function [30]. At the end of the experiment, TEWL in the DNCB group (74.3 g/m^2^/h) was 2.5 times that of untreated controls (31.7 g/m^2^/h), whereas animals in the DNCB/LCE group (58.9 g/m^2^/h) had a TEWL 21% lower than those in the DNCB group (Figure 1B). DNCB treatment reduced skin moisture content by 52.7% versus untreated controls (27.3% vs. 52.5%), and LCE application enhanced skin moisture content by 25% (to 34.2%) versus the DNCB group (Figure 1C). Skin surface pH was increased by DNCB (to 7.1) and reduced by LCE treatment (to 6.7) (Figure 1D). Next, levels of IgE, interleukin 4 (IL-4), and TSLP were evaluated in sera. The DNCB-induced AD group showed an increase in serum total IgE (203.0 ng/mL), IL-4 (25.8 pg/mL), and TSLP (328.0 pg/mL), but serum total IgE (142.9 ng/mL), IL-4 (18.5 pg/mL), and TSLP (234.2 pg/mL) were significantly lower in the DNCB/LCE group (Figure 1E–G).

### 2.2. Effects of LCE on Histological Changes in DNCB-Treated Mice

To investigate the effects of LCE on DNCB-induced hyperkeratosis and immune cell infiltration, we assessed epidermal and dermal morphological changes in DNCB-treated mouse tissues. H&E staining revealed more hyperkeratosis (over-proliferation of keratinocytes in the SC layer) and more epidermal hardening in the DNCB group than in the DNCB/LCE group (Figure 2A,C). Numbers of infiltrated immune cells in lesion sites were determined by toluidine blue staining. In the DNCB group, the number of mast cells infiltrating dermis was greater than in the untreated control group, and LCE treatment reduced this increase (Figure 2B,D). To confirm the protective effect of LCE on the skin barrier, we evaluated the locations and extents of LEKTI, KLK5, DSC1, and filaggrin expressions in tissue sections. IHC showed that LEKTI was expressed in the SC layer in the untreated group, and that its expression was lower in the DNCB group but higher in DNCB/LCE group than in the DNCB group (Figure 2E,I and Appendix A). KLK5 is a serine proteinase and is inhibited by LEKTI. Its expression was markedly increased in the SC layer in the DNCB group but lower in the DNCB/LCE group than in the DNCB group. Furthermore, this DNCB-induced increase in KLK5 expression and its suppression by LCE corresponded to observed degrees of hyperkeratosis (Figure 2F,J and Appendix A). DSC1 is a desmosomal protein that adheres to intercellular junctions and is degraded by KLK5. DSC1 was expressed in epidermis in the control group and reduced by DNCB, and this reduction was suppressed in the DNCB/LCE- group (Figure 2G,K). Under normal conditions, filaggrin is expressed in the SG and SC epidermal layers. DNCB reduced filaggrin expression in epidermis, but its expression was similar in the DNCB/LCE and control groups (Figure 2H,L). Collectively, these observations show that LCE can prevent hyperkeratosis and maintain moisturizing function by increasing the expression of LEKTI and inhibiting the activation and expression of KLK5, thereby preserving bonds between keratinocytes.

### 2.3. HPLC-PDA Analysis

Five compounds were identified in LCE by comparing retention times and PDA spectra with reference compounds. The major peak (peak 1, Figure 3A) was identified as diosmin (retention time (*t*_R_) 11.538 min and UV absorption maxima (*λ*_max_) at 250.8 nm and 346.2 nm). Peak 2 was identified as linarin (*t*_R_ 13.153 min, *λ*_max_ 267.4/331.8 nm), peak 3 as tomentin (*t*_R_ 13.612 min, *λ*_max_ 324.6 nm), peak 4 as diosmetin (*t*_R_ 15.778 min, *λ*_max_ 252.0/347.4 nm), and Peak 5 as 6,8-dimethoxycoumarin (*t*_R_ 17.320 min, *λ*_max_ 254.4/325.8 nm). Thus, three flavonoids (diosmin, linarin, and diosmetin) and two coumarins (tomentin and 6,8-dimethoxycoumarin) were detected (Figure 3B,C). In the previous experiment, HPLC-PDA results of *L. chinensis* water extract revealed that diosmetin was one of the marker compounds [31]. When we evaluated the effects of these five components by assessing the transcriptional activity of *SPINK5* using a dual-luciferase assay, only diosmetin increased its transcriptional activity (Figure 3D); this showed that diosmetin was responsible for the increased expression of *SPINK5* induced by LCE.

### 2.4. Regulatory Effect of Diosmetin on SPINK5/LEKTI Downstream Signaling

To determine the effects of diosmetin on SPINK5/LEKTI regulation and its downstream signaling, we estimated the gene expression levels of *SPINK5, KLK5*, and *KLK7* and protein expression levels of LEKTI, KLK5, KLK7, PAR2, and TSLP in UVB-irradiated HaCaT keratinocytes. Diosmetin at 3, 10, or 30 μM enhanced the transcriptional activation of SPINK5 promoter dose dependently, as revealed by a reporter gene assay using dual-luciferase in CV-1 cells (Figure 4A). *SPINK5* mRNA levels were reduced by UVB irradiation in HaCaT cells, and this was restored in the presence of diosmetin (Figure 4B).

On the other hand, *KLK5* mRNA and protein levels were increased by UVB, and these increases were significantly prevented by diosmetin (Figure 4C,D). Furthermore, UVB-induced LEKTI deficiency caused the over-activations of KLK5 and KLK7, and this over-activation of KLK5 increased PAR2 and TSLP protein levels. LEKTI protein levels were diminished by UVB, but this was prevented by diosmetin. In addition, UVB exposure-induced increases in the protein levels of KLK5, KLK7, PAR2, and TSLP (components of the signaling cascade downstream of LEKTI) were prevented by diosmetin (Figure 4E,F). In summary, these results show that diosmetin can improve skin barrier function by regulating LEKTI and its downstream signaling at the cellular level.

### 2.5. Effects of Diosmetin on the Skin Barrier in the DNCB Mouse Model

DNCB-induced skin redness, dryness, and hyperkeratosis were reduced by topical diosmetin (Figure 5A). Epidermal hyperplagia and immune cell infiltration were investigated using DNCB-lesioned mouse tissues. H&E staining detected hyperplasia in epidermis and hyperkeratosis in the SC layer in the DNCB group, and diosmetin application reduced these changes (Figure 5B,D). In addition, toluidine blue staining showed that numbers of infiltrating immune cells were lower in the diosmetin group than in the DNCB group (Figure 5C,D). Furthermore, detrimental TEWL, water content, and pH changes induced by DNCB were prevented by diosmetin treatment, and DNCB-induced increases in serum levels of IgE, IL-4, and TSLP were all significantly suppressed by diosmetin, indicating improved skin barrier function (Figure 5D). Immunohistochemical analysis demonstrated DNCB-induced reductions in LEKTI levels were suppressed by diosmetin (Figure 5E and Appendix A), as were DNCB-induced increases in KLK5 levels in epidermis, especially in the outermost layer of the SC (Figure 5F and Appendix A). Interestingly, DNCB-induced reductions in DCS1 levels (a keratinocyte adhesion molecule) were reduced by diosmetin (Figure 5G). Also, diosmetin significantly increased filaggrin levels in the SC layer (Figure 5H). In summary, these findings are in good accordance with previously published data [32,33] and indicate that diosmetin can alleviate the pathophysiological and hematological changes associated with AD, prevent hyperkeratosis of the skin, and maintain hydration function by increasing LEKTI expression and inhibiting the activation and expression of KLK5.

## 3. Discussion

The symptoms of AD are generally considered to promote skin penetration by anti-gens and induce stress and cytokine signaling in keratinocytes due to premature SC de-tachment. The skin barrier is a critical defensive feature regulated by the proliferation and differentiation of epidermal keratinocytes and the exfoliation of keratinized cells of the SC layer. Keratinocyte desquamation is regulated by serine protease and its endogenous in-hibitor, SPINK5/LEKTI. LEKTI can diminish the activities of epidermal serine proteases such as KLK5, KLK6, KLK7, KLK13, and KLK14 in skin [34], and a mutation in the SPINK5 gene is characteristic of Netherton syndrome, which presents with serious dehydration, itching, and chronic skin inflammation. Individuals with this syndrome have impaired skin barrier functions and elevated serum IgE levels, are susceptible to bacterial infections and allergens, and present with AD-like skin symptoms [35,36]. Low LEKTI expressions result in KLK activation and enhanced cleavage of keratinocyte desmosomal proteins (e.g., DSG1, DSC1, corneodesmosin (CDSN)) at SG-SC junctions, which results in pathologic epidermal exfoliation [37]. In this study, we targeted the regulation of SPINK5/LEKTI expression as a means of restoring DNCB-induced skin barrier dysfunction.

To study the skin barrier function of SPINK5/LEKTI, we used a murine model of DNCB-induced AD in which SPINK5/LEKTI expression was reduced. Sensitization and induction of allergic inflammation by dorsal application of DNCB closely mimics AD and exhibits skin barrier dysfunctions such as increased water loss and dry skin [38]. We ob-served hyperkeratosis, erythema, and redness in DNCB-induced lesions, and that these symptoms were alleviated by LCE. In addition, DNCB-induced disruptions of skin TEWL and hydration and skin surface pH and DNCB-induced increases in the serum levels IgE, IL-4, and TSLP were prevented by LCE. Histologic examination also showed that LCE prevented DNCB-induced skin barrier dysfunction and reduced DNCB-induced hyperkeratosis, epidermal thickness, and mast cell infiltration into dermal lesions. In normal skin, LEKTI is located in lamellar bodies at the junction between SC and SG layers and is separated from KLK5 and KLK7 vesicles in the upper SG layer [39]. Our histological analysis showed that DNCB reduced LEKTI, DSC1, and filaggrin levels but increased KLK5 levels in the SC layer, and that LCE treatment prevented these changes in LEKTI and KLK5 levels. Furthermore, reductions in KLK5 were paralleled by increases in DSC1 (a corneodesmosomal protein). In addition, LCE prevented DNCB-induced reductions in filaggrin levels. SPINK5 gene expression in keratinocytes is reported to be putatively regulated by transcription factors targeting the promoter regions −932/−837, −676/−532, and −318/−146 [40]. However, it is not yet known which factors play an essential role in SPINK5 gene expression. It is presumed that LCE increases SPINK5 gene expression by regulating transcription factors by acting on the promoter region. Taken together, these findings suggest that topical LCE treatment might improve the symptoms of AD.

Diosmetin has an anti-inflammatory effect by reducing IL-4, IL-1β, and TNFα in a DNCB model [32,33]. However, the improvement effect of atopic dermatitis through the skin barrier function related to SPINK5 has not been studied. When we evaluated the effects of the five components in LCE on the transcriptional activity of SPINK5, we found that diosmetin had the greatest effect. In a subsequent experiment, we verified that diosmetin had a skin barrier-improving effect in UVB-irradiated keratinocytes and a mouse model of impaired skin barrier function. Exposure of skin to UVB has been shown to drive the desquamation of epidermal keratinocytes [41]. To identify factors related to UVB-induced SC desquamation, we examined the expression levels of several molecules in HaCaT keratinocytes altered by UVB exposure. UVB irradiation resulted in a reduction in *SPINK5* and increases in *KLK5* and *KLK7* expressions at the mRNA and protein levels. Furthermore, reduced expression of SPINK5/LEKTI induced by UVB increased KLK5 activity, and this enhanced the expressions of PAR2 and TSLP, which can cause abnormalities in inflammatory and immune responses and eventually damage skin barrier function [42]. KLK5 plays the main essential roles in skin desquamation. KLK5 self-activates and also activates KLK7 and KLK14 [43]. Discrepancies between protein levels and mRNA of *KLK7* require further study considering the order and time of expression. Diosmetin reduced both *KLK5* and KLK*7* mRNA expression. However, the protein increased, which is thought to be because KLK5 is increased by UVB to activate KLK7 and KLK14. Therefore, more consideration is required for UVB and diosmetin treatment time. In addition, the increased expressions of KLK5 caused the breakdown of bonds between corneodesmosomal proteins, such as DSC1 and DSG1, thereby further weakening the skin barrier. However, treatment with diosmetin increased SPINK5 expression and inhibited the over-activations of KLK5, KLK7, and PAR2 and the expression of TSLP. These results show that the regulation of SPINK5/LEKTI may play an important role in disease-associated skin barrier disruption by modifying the protease cascade leading to skin inflammation [44,45]. According to our results, diosmetin, an active ingredient in LCE, increases SPINK5 expression and prevents the formation of DNCB-induced AD-like lesions with the same efficacy as LCE.

In summary, this study shows that LCE and its active compound diosmetin can prevent impaired skin barrier function by preventing the DNCB- or UVB-induced downregulations of SPINK5/LEKTI in vivo and in vitro, respectively, and thus, can disrupt the pro-tease cascade in atopic skin.

## 4. Materials and Methods

### 4.1. Plant and Extraction

Whole parts of *L. chinensis* used in this experiment were purchased from a local market (Deokhyeondang, Seoul, Korea) and were authenticated by one of the authors (S.N. Kim). A voucher specimen (KIST-2018-LC1010) was deposited at the herbarium of the Natural Products Research Institute of the Korea Institute of Science and Technology (KIST). Dried *L. chinensis* (100 g) was extracted three times using an ultrasonic device (Bandelin Sonorex, Berlin, Germany) with 95% ethanol (EtOH) for 3 h, filtered, and evaporated under vacuum at 30 °C to yield *L. chinensis* extract (LCE, 9.2 g).

### 4.2. Chemicals

6,8-Dimethoxycoumarin, tomentin, and diosmetin were isolated from *L. chinensis* by column chromatography and used as reference standards. Their structures were confirmed by nuclear magnetic resonance (NMR) [46,47,48], and their purities by HPLC (≥98.0%). Other reference standards, namely, diosmin (≥98.0%) and linarin (≥98.0%), were purchased from ChemFaces Biochemical Co., Ltd. (Wuhan, China). HPLC-grade water, acetonitrile, and methanol were purchased from Honeywell Burdick and Jackson (Muskegon, MI, USA). Formic acid was from DAEJUNG Chemicals & Metals Co., Ltd. (Siheung-si, Korea).

### 4.3. High-Performance Liquid Chromatography–Photodiode Array (HPLC–PDA) Conditions

The HPLC-PDA system consisted of a Waters Alliance e2695 separation module, a 2998 PDA detector (Waters Corporation, Milford, MA, USA), and a Waters SunFire^®^ C18 column (5 μm, 4.6 mm × 250 mm i.d.). The mobile phase consisted of 0.1% formic acid in water (eluent A) and 0.1% formic acid in acetonitrile (eluent B). Gradient elution was performed as follows: 0–3 min, 5–10% B; 3–23 min, 10–90% B; 23–25 min, isocratic 90% B. Column conditions were as follows: temperature, 40 ± 2 °C; injection volume, 10 µL; and flow rate, 0.7 mL/min. Detection was performed at 340 nm, and spectra were obtained from 210 to 400 nm. Empower 3 chromatography data software (CDS) was used to collect and analyze chromatographic data.

### 4.4. Cell Culture

HaCaT (spontaneously immortalized human keratinocytes) were purchased from AddexBio (catalog no. T0020001; San Diego, CA, USA) and grown in Dulbecco’s modified essential medium (DMEM, Hyclone, Logan, UT, USA) supplemented with 10% fetal bovine serum and 100 units/mL of penicillin and 100 µg/mL of streptomycin (Hyclone) in a humidified 5% CO_2_ atmosphere at 37 °C. CV-1 cells (monkey kidney cell line) were acquired from the Korean Cell Line Bank (KCLB No. 22256, Seoul, Korea) and grown in DMEM without sodium pyruvate (Hyclone), containing 10% fetal bovine serum and 100 units/mL penicillin and 100 µg/mL streptomycin (Hyclone) in a humidified 5% CO_2_ atmosphere at 37 °C.

### 4.5. UVB Irradiation

A bank of 6 Sankyo Denki G15T8E fluorescent UVB lamps (Sankyo Denki, Tokyo, Japan) with an emission range of 280–360 nm and peaking at 312 nm (UVA 30%; UVB 54%, UVC 0.2%) was used as the UVB source. Prior to UVB irradiation (15 mJ/cm^2^), HaCaT cells were pretreated with indicated samples, washed twice with phosphate-buffered saline (PBS), irradiated with UV light, and incubated in cell culture medium containing the same samples in a humidified 5% CO_2_ atmosphere for 24 h at 37 °C.

### 4.6. Real-Time Quantitative PCR (Q-PCR)

Total RNA was extracted from cells using RLT buffer (Qiagen, Hilden, Germany), according to the manufacturer’s instructions. cDNA was constructed from one μg of extracted RNA using a RevertAid first strand cDNA synthesis kit (ThermoFisher Scientific, Bremen, Germany). Real-time PCR was conducted using a 7500 Real-time PCR system (Applied Biosystems, Foster City, CA, USA) using SYBR1 Green (Power SYBR Green PCR Master Mix, Applied Biosystems). The primer sets used were as follows: *SPINK5*, forward: 5′-ATG AAG ATC AGG AAA TGT GCC A-3′ and reverse: 5′-GCC TTG GGA GCT CTT GCT AA-3′; *KLK5*, forward: 5′-TGT TCC AGG GGG TCA AAT CC-3′ and reverse: 5′-GGA GGA CCT TAG GGA AGT GC-3′; *KLK7*, forward: 5′-CTC AGT GGC AAT CAG CTC CA-3′ and reverse: 5′-AAA ACG CCT GCA ATG GTG AC-3′; glyceraldehyde-3-phosphate dehydrogenase (*GAPDH*), forward: 5′-AGG GCT GCT TTT AAC TCT GGT-3′ and reverse: 5′-CCC CAC TTG ATT TTG GAG GGA-3′. The PCR settings were as follows: initial denaturation for 15 s at 95 °C followed by 35 amplification cycles of 95 °C for 3 s and 60 °C for 31 s. Melting curve analysis was performed in the temperature range 60–95 °C. Relative gene expression levels were calculated by normalizing versus *GAPDH*.

### 4.7. Luciferase Reporter Gene Assay

CV-1 cells were inoculated into 24-well plates and cultured for 24 h before transfection. Cells were cotransfected with SPINK5pro-Luc reporter plasmid (constructed in a previous study [28]) and a control plasmid pRL-SV-40 using TransFast reagent (Promega, Madison, WI, USA) and cultured for 24 h. Then, the cells were treated with LCE or LCE components at the indicated concentrations and lysed after 24 h using a passive lysis buffer (Promega). Luciferase assays were performed with the Dual-Luciferase^®^ Reporter Assay System (Promega), according to the manufacturer’s instructions. Relative luciferase activities were calculated with respect to Renilla luciferase activities.

### 4.8. Immunoblotting

Protein expression levels of human SPINK5, KLK5, KLK7, PAR2, and TSLP were evaluated by immunoblotting. Briefly, cells or tissues were homogenized in RIPA buffer (Bioprince, Chuncheon, Korea) containing Phosphatase Inhibitor Cocktail 2,3 (Sigma Aldrich, St. Louis, MO, USA). Cell or tissue lysates were centrifuged at 13,000 rpm for 15 min at 4 °C, and supernatants were harvested. After measuring protein contents, 20 mg of proteins were separated by SDS-PAGE (sodium dodecyl sulfate polyacrylamide gel electrophoresis) and then transferred to polyvinylidene fluoride (PVDF) membranes. Blots were incubated with primary antibodies against LEKTI, KLK5, PAR2, TSLP (Abcam, Cambridge, UK), KLK7 (Santa Cruz Biotechnology, Santa Cruz, CA, USA), and GAPDH (Sigma Aldrich) and then with horseradish-peroxidase-conjugated secondary antibodies (Cell Signaling Technology, Denver, MA, USA). Visualization was performed using a SuperSignal^TM^ West Femto ECL kit (ThermoFisher Scientific). Band densities were calculated with ImageJ software (version. 1.5.2) and normalized with respect to GAPDH.

### 4.9. Animal Experiments

Female SKH-1 hairless mice (6 weeks old) were obtained from the Orientbio animal facility (Sungnam, Korea). Animals were maintained in a temperature- and humidity-controlled facility (23 °C ± 2 °C; 55% ± 5% RH) under a 12 h-light/dark cycle and supplied with standard animal chow and water ad libitum. All animal experiments were conducted in accordance with the Guide for the Care and Use of Laboratory Animals published by the National Institutes of Health (NIH publication no. 85-23, revised 1996) and approved by the Institutional Animal Care and Use Committee of KIST (Certification no. KIST-2020-001). Sensitization was performed by applying DNCB (200 μL, 1%) daily to mouse dorsal skin for 1 week, and then mice were challenged three times weekly for 2 weeks with 200 µL of 0.1% DNCB. During the challenge period, 200 μL of vehicle (propylene glycol: EtOH = 7:3), LCE (1%), or diosmetin (0.5%) was topically administered 4 h before and after each DNCB administration. Skin TEWL, hydration, and pH were measured weekly from 4 h after final challenge. Mice were sacrificed 3 weeks after the first DNCB application. TEWL (Aquaflux, Biox Systems, London, UK), hydration (SKIN-O-MAT, Cosmomed, Ruhr, Germany), and pH were measured at three sections weekly.

### 4.10. Enzyme-Linked Immunosorbent Assay

At the end of the experiment, blood samples were collected. Serum samples were obtained using a microcentrifuge (8000 rpm, 15 min). Total IgE, IL-4, and TSLP levels in serum were measured using Mouse IgE or Mouse IL-4 ELISA kits (Ready-SET-Go! eBioscience, San Diego, CA, USA) or a Mouse TSLP DuoSet ELISA kit (R&D Systems, Minneapolis, MN, USA), respectively.

### 4.11. Histological Examination

Dorsal skin (1.5 × 0.5 cm) was removed and fixed in 3.7% formaldehyde (Sigma-Aldrich) on white filter paper. Paraffin blocks were then prepared and sectioned at 4 μm. Sections were then stained with hematoxylin and eosin (H&E) to examine general morphologies or with toluidine blue to observe mast cells. Captured images and epidermal thickness were determined using ProgRes^®^ CapturePro application software (JENOPTIK laser, Jena, Germany) and viewed at ×200. Mast cell number and epidermal thickness were determined by randomly selecting three sites.

### 4.12. Immunohistochemistry

Deparaffinized 4 μm sections were immunohistochemically stained for LEKTI, KLK5, DSC1 (Abcam), and filaggrin (Biolegend). Goat Anti-Rabbit IgG H&L (Alexa Fluor^®^ 488, Abcam) was used as a secondary antibody. DAPI staining was performed using VECTASHIELD (Vector Laboratories, Burlingame, CA, USA). Images were captured using ProgRes^®^ CapturePro application software (JENOPTIK laser, Jena, Germany) and viewed at ×200. The tissue fluorescence intensity was quantified using ImageJ software (version. 1.5.2).

### 4.13. Statistical Analyses

Results are expressed as means ± standard deviations (SDs). The significances of intergroup differences were determined by ANOVA (one-way analysis of variance). *p*-values < 0.05 were considered statistically significant.

## 5. Conclusions

LCE or diosmetin increased the expression of SPINK5 in an in vitro UVB-induced cellular model and in an in vivo DNCB-induced AD murine model. Furthermore, LCE or diosmetin inhibited the effects of UVB or DNCB on events downstream of SPINK5/LEKTI. Furthermore, our data show that LCE or diosmetin can normalize serum IgE and IL-4 and skin mast infiltration immune responses. Taken together, our results suggest that LCE and diosmetin are good therapeutic candidates for the treatment of skin barrier-disrupting diseases such as Netherton syndrome or atopic dermatitis.

## Figures and Tables

**Figure 1 ijms-23-08687-f001:**
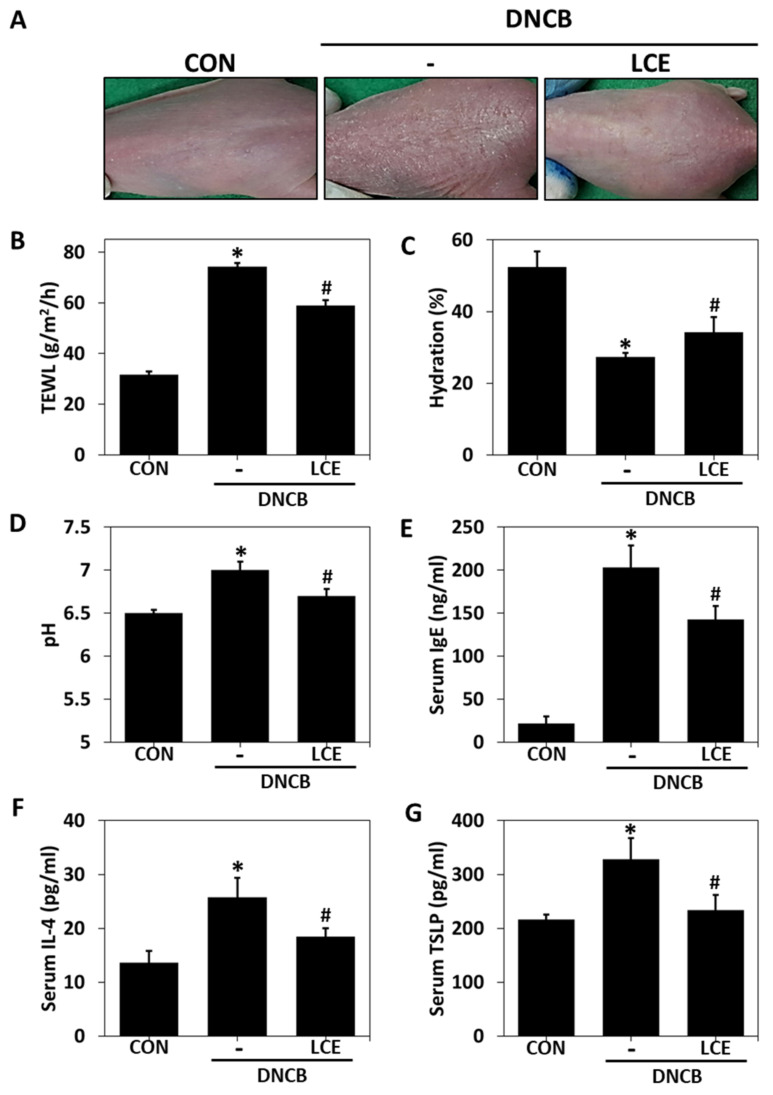
Effects of LCE on epidermal barrier function and serum factors associated with atopic symptoms in the DNCB-induced AD mouse model. (**A**) Changes in clinical features induced by LCE. (**B**) Transepidermal water loss (TEWL), (**C**) skin hydration, and (**D**) skin surface pH values were measured. (**E**) Serum total IgE, (**F**) IL-4, and (**G**) TSLP levels were evaluated by ELISA. Results are expressed as means ± SEMs (n = 5). * *p* < 0.05 vs. CON; # *p* < 0.05 vs. the DNCB-induced AD group. CON—untreated controls; DNCB—DNCB-induced AD group; DNCB-LCE—DNCB-induced AD and 1% LCE-treated group; LCE—*Lobelia chinensis* extract; IgE—immunoglobulin E; IL-4—interleukin 4; TSLP—thymic stromal lymphopoietin.

**Figure 2 ijms-23-08687-f002:**
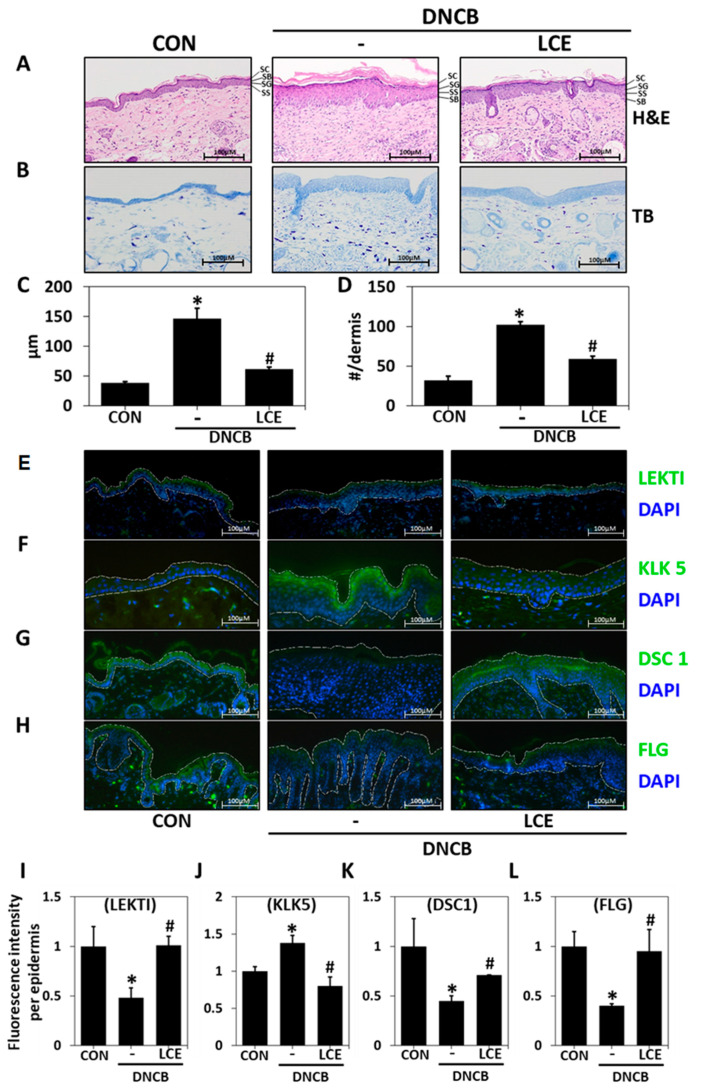
Histopathological analysis of the effects of LCE application in the DNCB-induced mouse model of AD. Tissues were excised, fixed in 10% formalin, embedded in paraffin, and sectioned. Tissue sections were stained with (**A**) hematoxylin and eosin (H&E), or (**B**) toluidine blue (TB). (**C**) Epidermal thicknesses were measured, and (**D**) mast cell numbers in dermis were counted. The changes in the protein levels of (**E**) LEKTI, (**F**) KLK5, (**G**) DSC1, and (**H**) FLG in tissue sections were detected by immunohistochemistry (original magnification 100×). White dashed lines demarcate the epidermis. (**I**–**L**) Protein fluorescence intensities were determined using ImageJ software. Results are expressed as means ± SDs (n = 5). * *p* < 0.05 vs. CON; # *p* < 0.05 vs. the DNCB-induced AD group. CON—untreated controls; DNCB—DNCB-induced AD group; DNCB-LCE—DNCB-induced AD and 1% LCE-treated group. LCE—*Lobelia chinensis* extract; LEKTI—lympho-epithelial Kazal-type-related inhibitor; KLK5—kallikrein 5; DSC1—desmocollin 1; FLG—filaggrin. SC—stratum corneum; SG—stratum granulosum; SS—stratum spinosum; SB—stratum basale.

**Figure 3 ijms-23-08687-f003:**
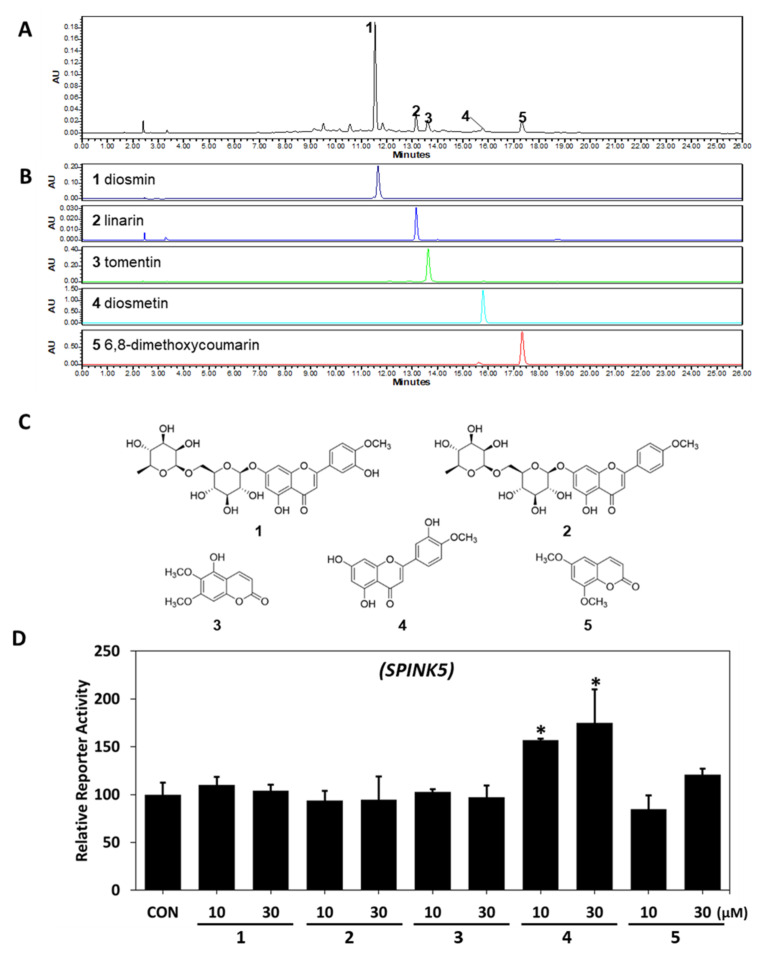
HPLC-PDA chromatograms of LCE and the effects of each of the five components of LCE on SPINK5. HPLC-PDA (**A**) chromatograms of LCE and (**B**) standard samples at 340 nm and (**C**) chemical structures. Peaks: (**1**) diosmin; (**2**) linarin; (**3**) tomentin; (**4**) diosmetin; (**5**) 6,8-dimethoxycoumarin. (**D**) Effects of the five components on *SPINK5* reporter activity were assessed in CV-1 cells co-transfected with a vector containing a *SPINK5* promoter and reference universal promoter (SV40). Results are expressed as the means ± SDs of three independent experiments. * *p* < 0.05 vs. CON (untreated controls). LCE—*Lobelia chinensis* extract; SPINK5—serine protease inhibitor Kazal type 5.

**Figure 4 ijms-23-08687-f004:**
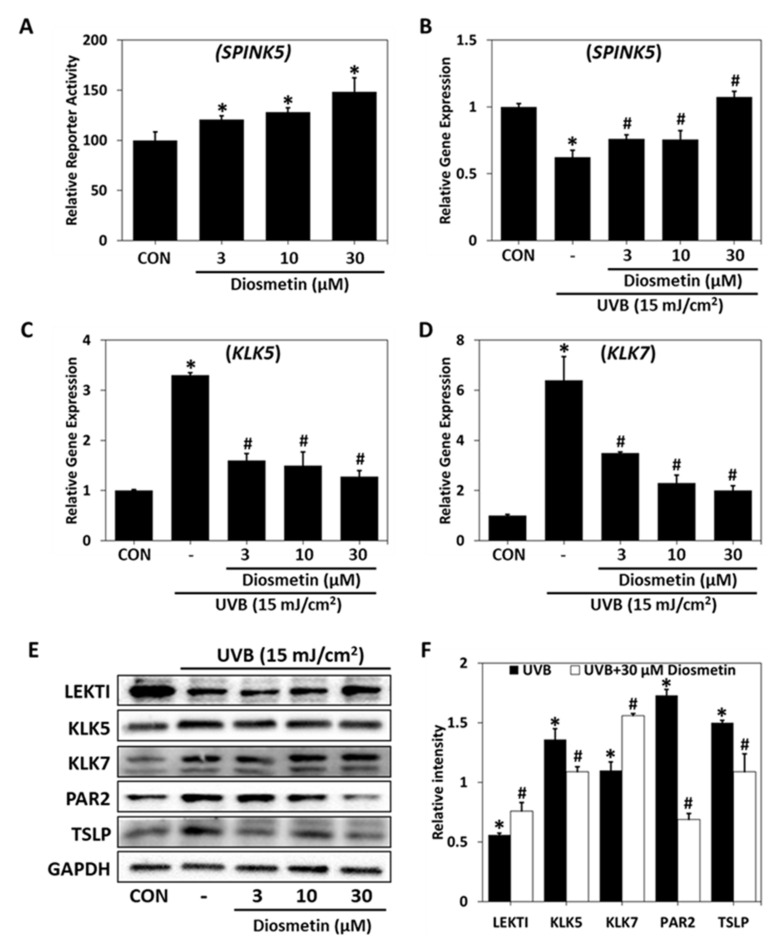
Regulatory effects of diosmetin on signaling downstream of SPINK5. (**A**) Effects of diosmetin on SPINK5 relative reporter activity were investigated in CV-1 cells cotransfected with a vector containing a *SPINK5* promoter vector containing a reference universal promoter (SV40). * *p* < 0.05 vs. CON (untreated controls). Relative gene expressions of (**B**) *SPINK5*, (**C**) *KLK5*, and (**D**) *KLK7* were determined by PCR and normalized versus *GAPDH* in UVB-irradiated (15 mJ/cm^2^) HaCaT cells. (**E**) LEKTI levels and levels of proteins downstream of LEKT1 were determined by western blot in UVB-irradiated (15 mJ/cm^2^) HaCaT cells. (**F**) Protein band intensities in the UVB-irradiated group and in the UVB-irradiated and diosmetin (30 μM)-treated group are expressed with respect to those in the control group. Results are presented as the means ± SDs of three independent experiments. * *p* < 0.05 vs. CON; # *p* < 0.05 vs. the UVB-irradiated group. CON—non-treated controls; UVB—UVB-irradiated group; UVB-Diosmetin—UVB-irradiated and diosmetin-treated group. SPINK5—serine protease inhibitor Kazal type 5; KLK5—kallikrein 5; KLK7—kallikrein 7; LEKTI—lympho-epithelial Kazal-type-related inhibitor; PAR2—protease activated receptor 2; TSLP—thymic stromal lymphopoietin.

**Figure 5 ijms-23-08687-f005:**
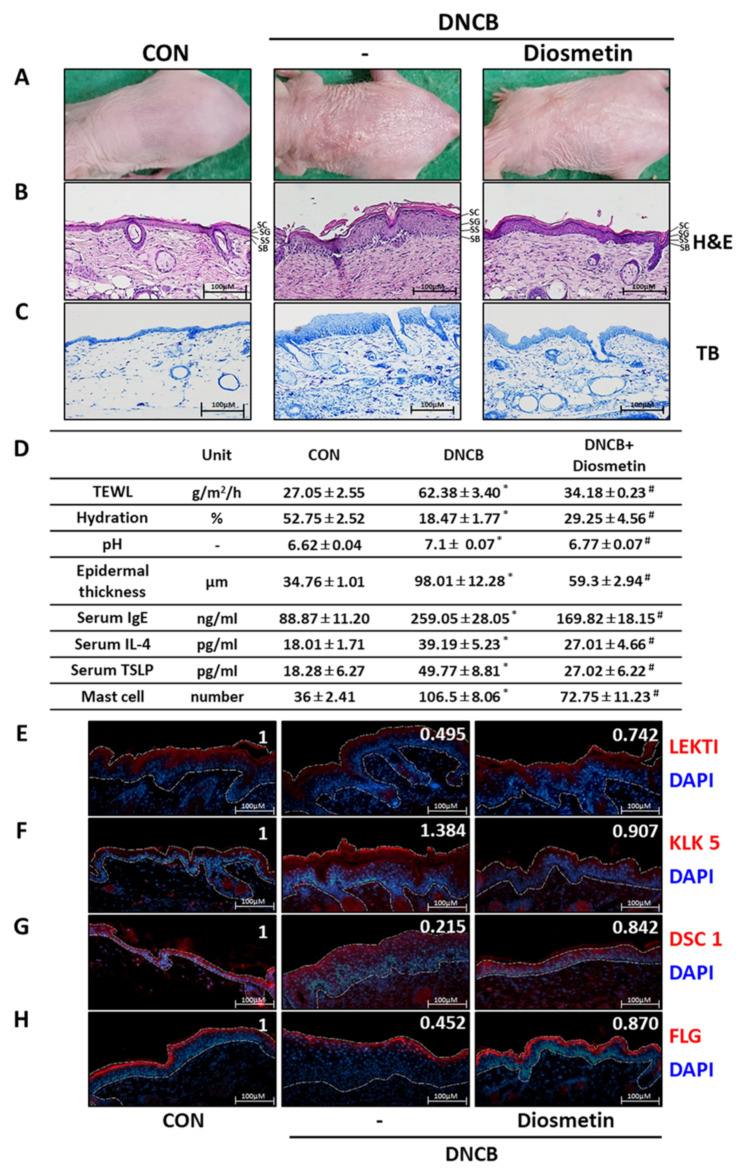
The effects of diosmetin on skin barrier function, serum factors, and histologic changes in the DNCB-induced AD model. (**A**) Clinical features of external skin after diosmetin treatment. Tissues were excised, fixed in 10% formalin solution, embedded in paraffin, and sectioned, and sections were stained with (**B**) hematoxylin and eosin (H&E) and (**C**) toluidine blue (TB). (**D**) Transepidermal water loss (TEWL), skin hydration, and skin surface pH were measured. Dorsal skin epidermal thickness was measured, and mast cell numbers in dermis were counted. Serum total IgE, IL-4, and TSLP levels were determined by ELISA. (**E**) LEKTI, (**F**) KLK5, (**G**) DSC1, and (**H**) FLG were detected in tissue sections by immunohistochemistry (original magnification 100×). White dashed lines demarcate the epidermis. Protein fluorescent intensities were determined using ImageJ software and are shown in the top right of the panel inset. Results are presented as means ± SEMs (n = 5). * *p* < 0.05 vs. CON; # *p* < 0.05 vs. the DNCB-induced AD group. CON—non-treated controls; DNCB—DNCB-induced AD group; DNCB-Diosmetin—DNCB-induced AD and 0.5% diosmetin-treated group. IgE—immunoglobulin E; IL-4—interleukin 4; TSLP—thymic stromal lymphopoietin; LEKTI—lympho-epithelial Kazal-type-related inhibitor; KLK5—kallikrein 5; DSC1—desmocollin 1; FLG—filaggrin. SC—stratum corneum; SG—stratum granulosum; SS—stratum spinosum; SB—stratum basale.

## Data Availability

All data generated or analyzed during this study are included in this manuscript.

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
