# Peer review of "Lobelia chinensis Extract and Its Active Compound, Diosmetin, Improve Atopic Dermatitis by Reinforcing Skin Barrier Function through SPINK5/LEKTI Regulation"

_ijms, 2022, doi:10.3390/ijms23158687_

Round 1
Reviewer 1 Report
In their manuscript “Lobelia chinensis extract and its active compound, diosmetin, 2 improve atopic dermatitis by reinforcing skin barrier function 3 through SPINK5/LEKTI regulation” Park et al. examined the impact of a plant extract on induced skin barrier defects in a mouse model. The authors have investigated both the whole extract (LCE) and the component diosmetin for their protective effect against induced skin barrier damage.
Apart from the fact that some relevant papers on the topic were not cited, the manuscript contains data that have already been published by the authors.
1. The authors provide HPLC-PDA analysis of LCE and have detected 5 components, of which they have identified diosmetin as the component that induces transcription of SPINK5 (p7, l177-187, Figure 3). Three of these 5 components including diosmetin had been published by Wang et al. in 2018 as compounds of Lobelia chinensis using a HPLC/Q-TOF MS (Wang et al., 2018, molecules, doi:10.3390/molecules23123258; not cited by the authors). The authors themselves also published a paper in 2021 in which they identified diosmetin as one of 4 marker compounds by HPLC-PDA (Jo et al., 2021, Applied Sciences, doi.org/10.3390/app112412080; not cited by the authors).
2. Lee et al. analyzed the impact of diosmetin on the skin in a DNCB-mouse model of AD. Although these authors more focused on inflammation, they have already demonstrated that diosmetin reduces the epidermal thickness and the mast cell numbers in the DNCB model as shown by the authors in Fig. 5 A-D (Lee et al., 2020, International Pharmacology, doi.org/10.1016/j.intimp.2020.107046 ; not cited by the authors). Much more serious, however, is that the authors themselves have already published most of the data shown in Fig. 5 in a paper of their own using exactly the same model (Park et al., 2020, Biomolecules & Therapeutics, doi.org/10.4062/biomolther.2020.135; not cited by the authors).
Author Response
Response to Reviewer 1 Comments
Dear Reviewer,
Thank you for your valuable comments.
We answered point-by point the reviewers’ specific comments and all changes are marked by highlighted in red color in the revised manuscripts.
Reviewer’s comments)
In their manuscript “Lobelia chinensis extract and its active compound, diosmetin, 2 improve atopic dermatitis by reinforcing skin barrier function 3 through SPINK5/LEKTI regulation” Park et al. examined the impact of a plant extract on induced skin barrier defects in a mouse model. The authors have investigated both the whole extract (LCE) and the component diosmetin for their protective effect against induced skin barrier damage.
Apart from the fact that some relevant papers on the topic were not cited, the manuscript contains data that have already been published by the authors.
- The authors provide HPLC-PDA analysis of LCE and have detected 5 components, of which they have identified diosmetin as the component that induces transcription of SPINK5 (p7, l177-187, Figure 3). Three of these 5 components including diosmetin had been published by Wang et al. in 2018 as compounds of Lobelia chinensis using a HPLC/Q-TOF MS (Wang et al., 2018, molecules, doi:10.3390/molecules23123258; not cited by the authors). The authors themselves also published a paper in 2021 in which they identified diosmetin as one of 4 marker compounds by HPLC-PDA (Jo et al., 2021, Applied Sciences, doi.org/10.3390/app112412080; not cited by the authors).
Answer) As your comment, additional components and references related to L. chinensis have added to the introduction (Wang et al., 2018, molecules, doi:10.3390/molecules23123258) (p3, l96). The published paper in 2021 (Jo et al., 2021, Applied Sciences, doi.org/10.3390/app112412080) is about the ingredients for water extract of L. chinensis, and we added information about the composition of L, chinensis water extract to the result 2.3 part (p7, l188-190).
- 2. Lee et al. analyzed the impact of diosmetin on the skin in a DNCB-mouse model of AD. Although these authors more focused on inflammation, they have already demonstrated that diosmetin reduces the epidermal thickness and the mast cell numbers in the DNCB model as shown by the authors in Fig. 5 A-D (Lee et al., 2020, International Pharmacology, doi.org/10.1016/j.intimp.2020.107046 ; not cited by the authors). Much more serious, however, is that the authors themselves have already published most of the data shown in Fig. 5 in a paper of their own using exactly the same model (Park et al., 2020, Biomolecules & Therapeutics, doi.org/10.4062/biomolther.2020.135; not citedby the authors).
Answer) In this thesis, we argued that topical treatment of diosmetin in the DNCB model has an atopic improvement effect due to the enhancement of the skin barrier function related to SPINK5. The mentioned paper (Lee et al., 2020, Park et al., 2020) attempted to improve the inflammatory response of atopic dermatitis with oral administration of diosmetin and diosmin in the DNCB model. In addition, it is claimed that diosmetin treatment improves atopic dermatitis through the anti-inflammatory reaction such as IL-4 decrease. However, our experimental model aims to improve atopic dermatitis via skin barrier enhancement in topical applications of diosmetin. Even with the same DNCB model, we claim the diosmetin topical application model focuses on SPINK5-related skin barrier function.
The previously mentioned our papers have added to the discussion part (p12, l309-311).
Diosmetin has an anti-inflammatory effect by reducing IL-4, IL-1β and TNFα in DNCB model. However, the improvement effect of atopic dermatitis through the skin barrier function related to SPINK5 has not been studied.

Reviewer 2 Report
Atopic dermatitis (AD) is common worldwide. People of all ages may suffer from this condition. Symptoms range from excessively dry, itchy skin to painful, itchy rashes that cause sleepless nights and interfere with everyday life.
|
No-June Park and colleagues report in their manuscript titled “Lobelia chinensis extract and its active compound, diosmetin, improve atopic dermatitis by reinforcing skin barrier function through SPINK5/LEKTI regulation” some data indicating that Lobelia chinensis extract (LCE) and especially its flavonoid diosmetin increased the gene expression of SPINK5 in an and in vitro cell line model and in vivo DNCB-induced murine model. Furthermore, LCE normalized serum IgE and IL-4 level and skin mast infiltration immune responses. Their results can suggest that LCE is new therapeutic candidate for the treatment of skin with AD or Netherton syndrome. |
The concept, the performance and the interpretation of the experiments are convincing. The used methods prove author’s expertise in molecular and cell biology to perform this study. However, the description of the methods and the discussion of the experimental results are chaotic and requires refinement.
Minor points:
1. I am not a specialist in English, but sometimes the test is incomprehensible and incorrect, e.g. TEWL in the DNCB group (74.3 g/m2/h) was 2.5 times that of untreated controls (31.7 g/m2/h); Regulatory effect on SPINK5 of LCE.
2. I recommend the unify the correct format of human gene symbols in the whole article. The symbols of the human genes are written in italics and in capital letters.
33. Please complete the description of the immunofluorescence photos: Fig.. 2.EFGH, no scale, color description I understand blue DAPI, green tested protein ??, the marked zone is the epidermis???? The same problem with Fig. 5 EFGH.
44. There is no precise description of when LCE was used in in vitro and in vivo studies before UVB exposure. How long after UVB irradiation were the experiments performed?
55. Fig. 5 D There are no statistically significant differences in table.
66. In the methods, the reference gene is GAPDH and in the figure 4 description it is actin. Please specify.
77. In the discussion there is no description of the potential mechanism in which LCE can regulate the level of gene expression of SPINK5.
88. In the discussion, please comment on the discrepancy in the level of mRNA and KLK7 protein after exposure to UVB radiation and diosmetin Fig. 4 D and DF (also an error in the numbering of figures DD)
Author Response
Response to Reviewer 2 Comments
Dear Reviewer,
Thank you for your valuable comments.
We answered point-by point the reviewers’ specific comments and all changes are marked by highlighted in red color in the revised manuscripts.
Reviewer’s comments)
Atopic dermatitis (AD) is common worldwide. People of all ages may suffer from this condition. Symptoms range from excessively dry, itchy skin to painful, itchy rashes that cause sleepless nights and interfere with everyday life.
No-June Park and colleagues report in their manuscript titled “Lobelia chinensis extract and its active compound, diosmetin, improve atopic dermatitis by reinforcing skin barrier function through SPINK5/LEKTI regulation” some data indicating that Lobelia chinensis extract (LCE) and especially its flavonoid diosmetin increased the gene expression of SPINK5 in an and in vitro cell line model and in vivo DNCB-induced murine model. Furthermore, LCE normalized serum IgE and IL-4 level and skin mast infiltration immune responses. Their results can suggest that LCE is new therapeutic candidate for the treatment of skin with AD or Netherton syndrome.
The concept, the performance and the interpretation of the experiments are convincing. The used methods prove author’s expertise in molecular and cell biology to perform this study. However, the description of the methods and the discussion of the experimental results are chaotic and requires refinement.
During revision, the experimental method was revised in detail, and a discussion part about the experimental results was refined.
Minor points:
- I am not a specialist in English, but sometimes the test is incomprehensible and incorrect, e.g.TEWL in the DNCB group (74.3 g/m2/h) was 2.5 times that of untreated controls (31.7 g/m2/h); Regulatory effect on SPINK5 of LCE. ?
Answer) Before submitting our manuscript, English sentences were corrected by a native-speaking proofreading expert (NURISCO, Seoul, Korea).
- I recommend the unify the correct format of human gene symbols in the whole article. The symbols of the human genes are written in italics and in capital letters.
Answer) The human gene symbol in the article has modified with italic and capital letters. The corrected words are marked with red highlight in the texts, figure and figure legends. (p3, l115) (p7, l191, 193) (p8, l198,199,205,209,211) (p9, l223,224,225) (p13, l318,319,324,325) (p14, l392,394,395,397,401)
- Please complete the description of the immunofluorescence photos: Fig.. 2.EFGH, no scale, color description I understand blue DAPI, green tested protein ??, the marked zone is the epidermis???? The same problem with Fig. 5 EFGH.
Answer) The scale bar, target protein color and epidermis zone were modified in the Figure 2 and 5. Green and red indicate target proteins (LEKTI, KLK5, DSC1, FLG), and blue indicates DAPI (p6, Fig 2) (p11, Fig 5).
- There is no precise description of when LCE was used in in vitro and in vivo studies before UVB exposure. How long after UVB irradiation were the experiments performed?
Answer) A more detailed method for UVB irradiation and sample treating procedures were added to M&M 4.5 section (p14, l382-385).
- Fig. 5 D There are no statistically significant differences in table.
Answer) We added statistically significant differences in Fig. 5D table (p11, Fig 5D).
- In the methods, the reference gene is GAPDH and in the figure 4 description it is actin. Please specify.
Answer) In the figure 4 description section, actin is mislabeled. So we corrected it to GAPDH (p9, l225).
- In the discussion there is no description of the potential mechanism in which LCE can regulate the level of gene expressionof SPINK5.
Answer) We added reference about SPINK5 gene promoter regulation (https://doi.org/10.1007/s12199-014-0393-7) in the discussion section. And following sentences were added in the text (p12, l303-307).
SPINK5 gene expression in keratinocytes is reported to be putatively regulated by transcription factors targeting the promoter regions −932/−837, −676/−532 and −318/−146. However, it is not yet known which factors play an essential role in SPINK5 gene expression. It is presumed that LCE increases SPINK5 gene expression by regulating transcription factors by acting on the promoter region.
- In the discussion, please comment on the discrepancy in the level of mRNA and KLK7 protein after exposure to UVB radiation and diosmetin Fig. 4 D and DF(also an error in the numbering of figures DD)
Answer) KLK5 play the main roles in skin desquamation. KLK5 self-activates and also activates KLK7,14 (Kishibe, M., Physiological and pathological roles of kallikrein-related peptidases in the epidermis. Journal of Dermatological Science, 2019. 95(2): p. 50-55). Discrepancies between protein levels and mRNA of KLK7 require further study considering the order and time of expression. Diosmetin reduced both KLK5,7 mRNA expression. However, the protein increased, which is thought to be because KLK5 is increased by UVB to activate KLK7,14. Therefore, more consideration is required for UVB and diosmetin treatment time (p13, l322-328).
The number error in Fig 4 was corrected (p9, Fig 4 D E F).

Round 2
Reviewer 1 Report
I apologize that I was not aware of the different form of administration of diosmetin between the submitted manuscript and the earlier publications. However, a precise description of the mode of application for diosmetin is missing in the current paper (and would have to be supplemented). Only for LCE a topical treatment is mentioned in the text, but not in the material and methods section. Furthermore, it is not that surprising that local topical application of diosmetin in this model (Fig. 5 A-D) also leads to the same effects that had been published for systemic treatment (intraperitoneally in Lee et al. or orally in Park et al.), so the authors should comment that they partly confirm published data with a modified protocol.
Further comments:
1. The material and methods section must also be improved regarding the descriptions for the measurement/calculation of fluorescence intensity per epidermis, counting of mast cells, assessment of epidermal thickness, TEWEL, hydration, pH and the experimental set up.
- How much of the skin was examined (defined area? defined anatomical position?)
- Were lesional and non-lesional skin of the same mice compared?
- What is the control group in the mouse experiments, untreated or just sensitized but not challenged mice?
2. Does the application of LCE /diosmetin has any effect on the untreated skin?
3. What is the reason to use SKH-1 mice? I could not find a good characterization of the phenotype of the SKH-1 mouse in the literature, but an immunophenotype has been described for SKH2/J mice (Jackson laboratory). Can the authors exclude that the observed effects are not biased by the mutation underlying the SKH-1 mice?
4. What exactly does the specification “n=5” in the legends of Fig. 1, 2 and 5 mean? The authors should indicate how many times the experiments were repeated and how many mice were used per experiment and group in each case. Were all shown data from the same experiment? In this context the authors should comment on the fact that they appear to show the identical DNCB control in Fig 5A as in Fig 5A of their previous publication by Park et al. in J Gingseng Res 2019. Does this mean, that the data shown here and in the previous publication are from a joint experiment in which there were at least 4 groups (control, DNCB, DNCB + diosmetin; DNCB + compound K)? Or are the histologies in Fig. 5A of the current paper from mice from different experiments? In the latter case the authors should explain how they rule out interexperimental differences that might impact the results.
5. Since the authors assume a treatment option for AD – did they analyze, whether the treatment after challenge (without any further DNCB application) leads to a faster or better improvement of the skin with LCE/diosmetin administration than without?
6. As mentioned by the authors diosmetin has an anti-inflammatory effect in the DNCB model and reduces the expression of some cytokines including IL-4. Atopic dermatitis is thought to be a Th2 cell-mediated disease and Th2 cells are believed to be the main source for IL-4. Did the authors observe differences in T cells infiltration upon LCE/diosmetin in their model?
7. Fig. 2: The resolution of the immunofluorescence images is not convincing. The authors should confirm the data by RNA and/or Western blot. It would also be helpful to perform the staining for the different marker on consecutive slides and show the same section each time, e.g. for the epidermal thickness the shown examples differ very much within one condition (in particular for “DNCB -“), although the data in Fig. 2C and 5D state only small variations.
Author Response
Dear Reviewer,
Thank you for your valuable comments.
We answered point-by point the reviewers’ specific comments and all changes are marked by highlighted in blue color in the revised manuscripts.
Reviewer’s comments)
I apologize that I was not aware of the different form of administration of diosmetin between the submitted manuscript and the earlier publications. However, a precise description of the mode of application for diosmetin is missing in the current paper (and would have to be supplemented). Only for LCE a topical treatment is mentioned in the text, but not in the material and methods section. Furthermore, it is not that surprising that local topical application of diosmetin in this model (Fig. 5 A-D) also leads to the same effects that had been published for systemic treatment (intraperitoneally in Lee et al. or orally in Park et al.), so the authors should comment that they partly confirm published data with a modified protocol.
Answer) We added a method for topical application to M&M (p15, l437), and sentence that we partly confirm published data with a modified protocol to result (p10, l251-252).
Further comments:
- The material and methods section must also be improved regarding the descriptions for the measurement/calculation of fluorescence intensity per epidermis, counting of mast cells, assessment of epidermal thickness, TEWEL, hydration, pH and the experimental set up.
Answer) As your valuable comments, the description of fluorescence intensity measurement/calculation per epidermis, mast cell count, epidermal thickness assessment, TEWL, hydration, and pH were corrected in the M&M section. (p15, l440-441, l452-455, l462-463)
- How much of the skin was examined (defined area? defined anatomical position?)
Answer) TEWL, Hydration, and pH were measured at 3 sites per mouse. For histological analysis, tissues were collected in a size of 10 mm x 20 mm, and the collected tissues were stained and randomly selected at three locations to measure the epidermal thickness, the number of mast cells, and the fluorescence intensity.
- Were lesional and non-lesional skin of the same mice compared?
Answer) We couldn’t compared lesional and non-lesional skin of the same mice. Because DNCB, LCE or diosmetin were applied to the overall dorsal surface of the mouse.
- What is the control group in the mouse experiments, untreated or just sensitized but not challenged mice?
Answer) The control group in the mouse experiments is untreated mice.
- Does the application of LCE /diosmetin has any effect on the untreated skin?
Answer) We didn’t examine the effects of LCE/diosmetin on the untreated skin. But skin irritation of them was evaluated as an alternative skin irritation test method. HET-CAM assay as an alternative skin irritation test method was performed, and the result was negative for skin irritation. (Gilleron, L., et al. "Evaluation of the HET-CAM-TSA method as an alternative to the Draize eye irritation test." Toxicology in vitro 11(5): 641-644. 1997)
- What is the reason to use SKH-1 mice? I could not find a good characterization of the phenotype of the SKH-1 mouse in the literature, but an immunophenotype has been described for SKH2/J mice (Jackson laboratory). Can the authors exclude that the observed effects are not biased by the mutation underlying the SKH-1 mice?
Answer) Previous publication (The hairless mouse in skin research. J Dermatol Sci 53(1):10-18 2009 ) showed that SKH-1, unpigmented and immunocompetent mice, are the most widely used in dermatologic research. SKH-1 mice appear to be more sensitive to the immunosuppressive effects of hapten, especially suppressing Th1-mediated cellular immune responses. Therefore, SKH-1 is a suitable animal model for hapten induced Th2-mediated dermatitis.
According to The Jackson Laboratory, SKH2/J mice develop a hairless phenotype, hearing loss, and have a higher incidence of leukemia. One research article showed enhanced barrier function of pigmented skin with these mice model (J Inves Dermatol 134(9):2399-2407 2014). It seems that this model is not widely used by dermatologists.
- What exactly does the specification “n=5” in the legends of Fig. 1, 2 and 5 mean? The authors should indicate how many times the experiments were repeated and how many mice were used per experiment and group in each case. Were all shown data from the same experiment? In this context the authors should comment on the fact that they appear to show the identical DNCB control in Fig 5A as in Fig 5A of their previous publication by Park et al. in J Gingseng Res 2019. Does this mean, that the data shown here and in the previous publication are from a joint experiment in which there were at least 4 groups (control, DNCB, DNCB + diosmetin; DNCB + compound K)? Or are the histologies in Fig. 5A of the current paper from mice from different experiments? In the latter case the authors should explain how they rule out interexperimental differences that might impact the results. ?
Answer) "n=5" means 5 mice per test group. Fig. 1,2 and 5 are the results of animal experiments using 5 mice per group, and statistical analysis was performed based on the results of 5 animals. The results presented in Fig. 5 are different from the previous publications of Park et al. in JGR 2019. The data from the two papers may look similar. However, they were performed at different times. The animal approval number of the thesis published in 2019 is “Certification No. KIST-2016-011”, and the animal approval number of the thesis currently being written is “Certification No. KIST-2020-001”.
- Since the authors assume a treatment option for AD – did they analyze, whether the treatment after challenge (without any further DNCB application) leads to a faster or better improvement of the skin with LCE/diosmetin administration than without?
Answer) We didn’t analyze whether the treatment after challenge (without any further DNCB application) leads to a faster or better improvement of the skin with LCE/diosmetin administration than without, in this article.
DNCB treatment schedule is as follows.
IF there is no additional DNCB application after DNCB sensitization, the back skin of the mice is restored to the untreated skin as soon as DNCB treatment is stopped. Sorry but we cannot answer your question accurately. In addition, the purpose of this study is to aggravate the skin barrier with continuous DNCB boosting, and to confirm whether the deteriorated skin barrier is improved by LCE/diosmetin. So, we think it can be explained sufficiently with the data we are claiming now.
- As mentioned by the authors diosmetin has an anti-inflammatory effect in the DNCB model and reduces the expression of some cytokines including IL-4. Atopic dermatitis is thought to be a Th2 cell-mediated disease and Th2 cells are believed to be the main source for IL-4. Did the authors observe differences in T cells infiltration upon LCE/diosmetin in their model?
Answer) We didn’t observe differences in T cells infiltration upon LCE/diosmetin in this DNCB model. However, in the preliminary stage of this study, we checked the differentiation of Th0 cells into Th2 cells using EL4 cells. EL4 cells are T lymphoblasts that differentiate into Th2 cells by immunostimulatory chemicals (PMA + OP). The degree of differentiation was confirmed by the mRNA expression level of IL-4, a representative cytokine of Th2 cells. Although it was not possible to confirm T cell infiltration in mouse tissues, in vitro test results showed that LCE/diosmetin inhibited the differentiation of T lymphoblasts into Th2 cells.
- Fig. 2: The resolution of the immunofluorescence images is not convincing. The authors should confirm the data by RNA and/or Western blot. It would also be helpful to perform the staining for the different marker on consecutive slides and show the same section each time, e.g. for the epidermal thickness the shown examples differ very much within one condition (in particular for “DNCB -“), although the data in Fig. 2C and 5D state only small variations.
Answer) As your valuable comments, we performed Western blot for LEKTI and KLK5. Data from Western blot showed the same results as the immunofluorescence data in Figures 2 and 5. Therefore, we added it to the supplementary figure 2 (for LCE) and 3 (for diosmetin) (p5 l153 and l158, p10 l247 and l248).
We tried to perform the staining for the different marker on consecutive slides and show the same section each time. But we couldn’t get the good image on the same section of consecutive slides. So, we representatively showed the parts stained with the best quality among several images.
